# High-Throughput Identification of Adapters in Single-Read Sequencing Data

**DOI:** 10.3390/biom10060878

**Published:** 2020-06-08

**Authors:** Asan M.S.H. Mohideen, Steinar D. Johansen, Igor Babiak

**Affiliations:** Genomics Group, Faculty of Biosciences and Aquaculture, Nord University, P.O. Box 1490, 8049 Bodø, Norway; asan.m.haja-mohideen@nord.no (A.M.S.H.M.); steinar.d.johansen@nord.no (S.D.J.)

**Keywords:** 454 pyrosequencing, adapter oligonucleotides, adapter trimming, Illumina, Ion-Torrent, randomized adapters, single-read sequencing, small RNA sequencing, SOLiD

## Abstract

Sequencing datasets available in public repositories are already high in number, and their growth is exponential. Raw sequencing data files constitute a substantial portion of these data, and they need to be pre-processed for any downstream analyses. The removal of adapter sequences is the first essential step. Tools available for the automated detection of adapters in single-read sequencing protocol datasets have certain limitations. To explore these datasets, one needs to retrieve the information on adapter sequences from the methods sections of appropriate research articles. This can be time-consuming in metadata analyses. Moreover, not all research articles provide the information on adapter sequences. We have developed adapt_find, a tool that automates the process of adapter sequences identification in raw single-read sequencing datasets. We have verified adapt_find through testing a number of publicly available datasets. adapt_find secures a robust, reliable and high-throughput process across different sequencing technologies and various adapter designs. It does not need prior knowledge of the adapter sequences. We also produced associated tools: random_mer, for the detection of random N bases either on one or both termini of the reads, and fastqc_parser, for consolidating the results from FASTQC outputs. Together, this is a valuable tool set for metadata analyses on multiple sequencing datasets.

## 1. Introduction

Adapters are oligonucleotides ligated during the sequencing library preparation to the 3′ and sometimes 5′ ends of biological sequences [1]. The formed sequence templates are usually immobilized to a flow-cell or to beads, as in the case of Illumina or Ion Torrent sequencing technologies, respectively [2]. An adapter helps in the specific enrichment of adapter-ligated DNA during the PCR step. The size of the sequence of interest is referred to as the insert size [2]. Most sequencing machines are compatible to read insert sizes ranging from 35 to 1000 base pairs (bp) depending on the sequencing platform and instrumentation used [2,3]. When reads have insert sizes less than ~50 bp, the output from some sequencing platforms, such as Illumina or SOLiD, would also harbor the biological sequence along with the adapters [3]. In addition, some outputs from Illumina sequencing may also contain a four-letter barcode at the 5′ end. In the case of Ion Torrent and 454 platforms, both the 5′ and 3′ adapters are present.

In a default analytical pipeline, the adapter sequences have to be removed from the raw sequenced data prior to a downstream analysis of biological sequences. There are tools such as cutadapt that remove adapter sequences [3]. These tools, however, require input information about the adapter sequences and whether they are present at both the 5′ and 3′ ends. Thus, the adapter trimming step needs to be performed for each dataset individually, and the information on adapter specifications should be either retrieved from the source research papers, or if not available, obtained from authors upon request. If a large number of datasets have to be analyzed, this would require substantial resources and time to retrieve the original information on adapter specifications. In addition, some sequencing kits use randomized adapters protocol to reduce a bias resulting from the adapter sequence proximal to the ligation junction at both 5′ and 3′ ends [4]. These adapters have two to four randomized bases adjacent to the ligation junction.

A quality control check is a routine procedure applied before and after the adapter trimming. Tools such as FASTQC [5] produce output reports individually for each input sequencing file. When performing this operation on a large number of adapter- and quality-trimmed sequencing datasets, inspecting and extracting information from these individual files can be cumbersome. R package fastqcr can be used to parse the output of FASTQC [6]; however, a priori knowledge of R is needed to install the relevant dependencies and executing a batch of R commands. No relevant easy-to-use tools are available to execute and extract the relevant information in a single step.

There are tools available to identify adapter sequences, but most of them cannot be applied, or appear not accurate enough, for single-read sequencing data. BBMerge.sh from BBMap suite identifies adapters only in paired-end sequencing protocols [7]. The output of over-represented sequences from FASTQC analysis can be used to predict adapter sequences, but it is not always accurate since FastQC uses only the first 100,000 sequences to calculate the over-represented sequences [5]. fastp [8], Minion from Kraken package [9], and DNApi [10] can identify adapter sequences from a single-read sequencing file, but these tools are not absolutely reliable for analyzing single-read data. Moreover, DNApi and Minion can identify only 3′ end adapters and work only for sequencing files from the ILLUMINA sequencing platform.

Currently, there are no publicly available tools to perform automated, high-throughput and reliable pre-processing steps of adapter identification in single-read protocol sequencing datasets from any sequencing platform. We have developed adapt_find that accurately identifies adapters in single-read protocol datasets with no restrictions to the adapter type, sequencing platform technology, number of processed datasets, or input information on adapter sequences. In addition, we have produced associated tools: random_mer for the detection of random N bases either on one or both termini of reads, and fastqc_parser for consolidating results from FASTQC outputs. These tools are available online at https://github.com/asan-nasa/adapt_find.

## 2. Materials and Methods

### 2.1. Implementation and Dependencies

adapt_find is written in PYTHON 3.6.8 (Python Software Foundation, Beaverton, OR, USA), also compatible with PYTHON 2.7.5. It also requires PYTHON modules such as pandas [11], numpy, scipy, multiprocessing and other tools such as blatstn (2.7.1+) [12], cutadapt [3] and bowtie (1.1.2) [13]. blatstn, cutadapt, and bowtie have to be specified in the PATH in such a way that they can be invoked anywhere.

### 2.2. Adapter Types

Most often in the single-read sequencing, the type of adapter depends on the sequencing platform used. Adapters are ligated to either the 5′ or 3′ ends, or to both ends of an insert. The length of the 3′ end adapter can vary depending on the length of a biological sequence. The 5′ end adapters are usually of fixed length, although they can have insertions, deletions or substitutions. Under a randomized adapters protocol, two to four randomized nucleotides (random_mers) are added at the adapter’s ligation junction site to reduce the ligation bias [4]. In the case of 5′ end adapter trimming, the adapter sequence and any other sequence preceding the adapter are removed, whereas in the case of 3′ end adapter trimming, the adapter sequence and any sequence that follows it should be removed. Therefore, the efficient adapter trimming requires precise recognition of the start (5′ end or head end) sequence of a 3′ end adapter and the termination (3′ end or tail end) sequence of a 5′ adapter. As the 3′ end adapter length can vary, identification of the shortest length variant is sufficient for the adapter trimming (Figure 1).

In the Illumina sequencing technology, the output reads do not have 5′ adapters. However, the output data may contain indexing four- or six-letter barcodes, which are ligated to the 5′ end during the library preparation. There could also be four random nucleotides ligated to both the ends of the biological sequence (Figure 2). In the case of 454 pyrosequencing and Ion Torrent sequencing technologies, there are both 5′ end and 3′ end adapters. For Ion Torrent sequencing, there may be random_mers ligated to the 3′ end of the biological sequence (Figure 2). In the case of SOLiD sequencing, there is a standard 3′ adapter that is ligated to the 3′ end of biological sequence (Figure 2), and the file format is in the color space. Hence, different strategies have to be used for different sequencing platforms to identify adapter sequences.

### 2.3. Principles of the Method

#### 2.3.1. Illumina Platform

The common principle for finding adapters is described using the Illumina sequencing technology as an example (Figure 3). The read counts of unique sequences are created from a FASTQ file and sorted in a descending order. A quality check is performed to check whether the input FASTQ files are adapter-trimmed by examining the read length distribution of raw reads and adapter-trimmed files are not processed further. Untrimmed sequencing files with reads with a four- or six-letter barcode are expected to be of a constant length (the same sequence for all the tested reads-anchored adapter), whereas the 3′ end adapter can vary in length depending on the insert size. In the first step, a K-mer approach is used to identify adapter sequences in the first and the last six nucleotides on both the ends of the read sequences. This is done by extracting the first six nucleotides (k-mer) on both the ends of the reads from the top 25 most abundant raw reads in a given dataset. Then, the frequency of occurrence of a common sequence among k-mers is calculated. If no common sequence is found, the same sequence of steps is repeated with 5-mers and so forth until a 3–mer or until a common sequence is found. In the second step, a BLAST approach is used whereby 50 sequences are selected as a query and 200 sequences are selected as a subject from the collapsed raw read sequences. The selection of query and subject sequences is based on the total number of reads and the read count of the most abundant read sequence. The query and subject sequences are selected in such the way that they do not include homopolymers and low complexity sequences. Further validation is performed to filter out biological sequences similar between the query and the subject, and to retain putative adapter sequences. This is achieved by blasting the query sequences with the subject FASTA sequence using following parameters: (1) ungapped alignments and (2) two hits allowed for each query sequence. Common sequences from the BLAST output are taken as putative 3′ end adapters.

The rationale for allowing two hits per query sequence is that the common sequence between the query and the subject should be the adapter sequence. This holds true only if there are no similar biological sequences, such as length variants or sequence variants, between the query and the subject. If a similar biological sequence is present in both the subject and the query and if the length of the adapter is shorter than the attached biological sequence, the biological sequence rather than the adapter would be reported as a match in the BLAST output. By allowing two hits, the adapter sequence will be reported along with any biological similar sequence. Such biological sequences will be filtered out in the following steps based on the median length and the position of aligned sequences:median > 20: alignments with query or subject start sites less than 5 are removed. The rationale is that alignments with query or subject start sites are likely to have truncated adapters. Similarly, alignments with both query and subject start sites less than 15 are removed. The rationale is that in small RNA sequencing the minimum length of target sequence is usually 15. Therefore, the minimum start position of the adapter sequence should be 16, preferably in both query and subject sequences. Thereafter, aligned sequences with length less than or equal to 15 are discarded to remove non-adapter sequences. Sequencing datasets with read length greater than 100 nt will result in aligned sequences with median length greater than 60 nt. In such cases, alignments with both query start and subject start less than 20 nt and alignment length less than 60 nt are removed. Similar exceptions in the filter threshold are made if the difference between the median of aligned sequences and the median of raw read length is too high or too low.median between 10 and 20: similar filtering criteria as above are applied; in addition, aligned sequences with aligned length less than 10 are removed.median < 10: aligned sequences with query or subject start site less than or equal to 15 are removed.

The aligned sequences are further subjected to the quality control step (Figure 3) based on the length of the adapter sequences inferred from the median length of aligned sequences:median ≤ 10: Although not typical, in some studies the adapter sequence is short, such as in the dataset SRR578913 (Table 1). The presence of such short adapters can be ascertained through the median length of aligned sequences. For such datasets, only alignments less than 15 nt are retained. The retained sequences are sorted alphabetically, thus the length variants are grouped together as a data frame. This is done because, depending on the insert size (length of biological sequences), the length of the adapter sequences could vary. This would mean that the adapter sequences would exist as length variants in the pool of aligned sequences. The adapter sequence would ideally be the longest data frame among the list of data frames created. The list of data frames is sorted based on the size. If there are no biologically similar nucleotides preceding the adapter sequence between all the query and subject sequences, then there would be only one data frame. In such a case, the most common sequence is taken as the adapter. If there are more than two data frames, the two topmost data frames based on their size are taken. From each of these data frames, the sequence with the shortest length variant is taken. The common motif between the two sequences is taken as the 3′ end adapter.median > 10: The aligned sequences are grouped as data frames based on the query sequences. For each data frame, each of the aligned sequences longer than 20 nt is trimmed to 20 nt by removing bases from its 3′ end. This is because 3′ end adapter sequences are typically shorter than 21 nt. The position of the adapter sequence in the query sequence would remain constant, whereas it would change in the subject sequence. Hence, for a given query sequence, all the adapter sequence alignments should have the same query start site; this condition is not met if the matched sequence is not an adapter. Hence, alignments are extracted for each query sequence and the frequent query start position is computed. For each query sequence, only alignments that have the frequent query start position and start within nucleotide 1–3 after the frequent query start position are retained. This step filters out non-adapter sequences and takes into account the possible use of random_mers. After this step, the aligned sequences are counted to compute the most frequently occurring putative adapter sequences. If there is only one frequently occurring aligned sequence, this is taken as the adapter sequence. If more than two sequences are in the putative adapter list, the common sequence between the two sequences is taken as the adapter sequence.

#### 2.3.2. SOLiD Platform

The approach is slightly different for SOLiD sequencing datasets. SOLiD uses standard 3′ end adapter CGCCTTGGCCGTACAGCAG. However, any custom-made adapter can be predicted using the following workflow. The color space reads are converted to standard FASTQ format using a Perl script (https://gist.github.com/pcantalupo/9c30709fe802c96ea2b3) obtained from the github repository. This converted FASTQ is used only for finding the adapter sequence. The adapter sequences are identified from the converted FASTQ file similar to that of the Illumina adapters. Adapter trimming is performed on the original color space read files using cutadapt with the predicted adapter [3]. The adapter-trimmed file is then mapped against the reference color space genome index using bowtie with three mismatches to allow mapping for reads with non-templated additions, such as tRNA-derived fragments. Finally, the mapped reads are extracted from a Sequence Alignment Map (SAM) file and converted to the standard FASTQ file format. These latter two steps are optional.

#### 2.3.3. 454 and Ion Torrent Platforms

In 454 pyrosequencing and Ion Torrent sequencing datasets, both 5′ end and 3′ end adapters are retained in the raw data. Unlike Illumina sequencing, a fraction of the 5′ end adapters may occur in a degraded form at their 5′ ends. To identify the 5′ and 3′ adapters, BLAST is used similar to that of Illumina sequencing, but with some modifications. Here, 15 sequences are selected for the query file and 100 sequences for the subject file. Two hits per subject are used (one for the 5′ end adapter and the other for the 3′ end adapter). Then the aligned sequences are extracted, and the top two sequences are selected (5′ end and 3′ end adapters). The sequence that has frequent query start sites between 1 and 6 is the 5′ end adapter, and the other sequence is the 3′ end adapter.

### 2.4. Detection of Random 4-mers

The detection of randomized nucleotides, which are used in randomized adapter protocols, needs an additional procedure. The tool, random_mer, requires the same set of dependencies as adapt_find and uses the same blast approach. However, the reference genome of the species of interest is used as a subject for blasting. The top 750 sequences with length greater than or equal to 29 are used as query sequences, which are blasted against the reference genome with an ungapped alignment. The principle behind this approach is that when a read has random bases on both or either one of the ends, only the biological sequence of the read is aligned to the reference. Meanwhile, the part that has the random bases would not align to the reference genome. The presence or absence of random “N” bases can be computed by parsing the output parameters from BLAST such as the aligned sequence, query start, and query end position. The output csv file is parsed to check if all the aligned sequences have a common query start position. Similarly, the query end position is subtracted from the total length of the read to calculate the length of unaligned sequence at the 3′ end. Computing the most frequent start site will identify random_mers at the 5′ end. For example, if there is a random 4-mer at the 5′ end, most inserts will have the sequence start position at 5th nucleotide. Random_mers at the 3′ end are predicted by computing the length of an unaligned sequence at the 3′ end for each query sequence and calculating the most common length of all such unaligned sequences. The flowchart showing the steps involved in random_mer detection is shown in Figure 4.

### 2.5. Parsing the Outputs from FASTQC Tool

We developed fastqc_parser, a python script, to parse the outputs of the FASTQC tool from multiple sequencing files. The script runs FASTQC on input FASTQ files, extracts the output files, parses the output files to extract the results and outputs a csv file with filenames as row names and individual parameters as column names for individual files. The column names include filename, basic statistics, per base sequence quality, per sequence quality scores, per base sequence content, per sequence GC content, per base N content, sequence length distribution, sequence duplication levels, overrepresented sequences, and adapter content. Depending on user-defined input criteria “WARN” and/or “FAIL”, the output of problematic parameters is copied into a specified folder. The output includes images and the data corresponding to each filename.

## 3. Results

### 3.1. Usage

Running python “adapt_find.py –help” displays the usage options (Appendix A). The required parameter is the sequencing platform used for the input files, which can be “ILLUMINA”, “ION_TORRENT”, “SOLID” or “454” (Example: adapt_find.py ILLUMINA). The six optional arguments and their corresponding default values are listed in Appendix A. If the optional arguments are not specified, the default values apply. The options for the randomized adapters tool (random_mer) are the same, with an additional parameter for specifying the path to the reference genome.

### 3.2. Tests

Publicly available single-read sequencing data (811files) were downloaded from the NCBI SRA run selector for zebrafish (*Danio rerio*) and worm (*Caenorhabditis elegans*) using automated script and csv output. The summary of the file names, tissue sample/developmental stages, and other relevant information is provided in Appendix A (Illumina and 454 pyrosequencing), 3 (SOLiD), and 4 (Ion Torrent). The datasets were generated using different sequencing platforms, and they varied in the types and length of adapters. In addition, there were several datasets without any adapter sequences; adapt_find marked them as datasets with no adapter sequences.

We manually verified the adapter sequences in all the tested datasets either by checking the information available in NCBI SRA webpage or NCBI GEO accession webpage, or in the source research papers. The output of adapt_find analysis was fully concordant with the source data, and no discrepancies were found. The filename, the adapter sequence, the type of adapter and the mapping percentage for Illumina, SOLiD, Ion Torrent, and 454 technologies are given in Appendix A, respectively. adapt_find not only identified all the adapters reported in all the tested datasets (Table 1 and Appendix A), but it also retrieved the correct information on the adapters, which were erratically reported in the original source study (Table 2).

The fastqc_parser tool runs FASTQC tool on fastq files, unpacks the result files, parses the output files, and consolidates the output in a single csv file (Appendix A). This saves a lot of time when multiple datasets have to be analyzed. Users can select problematic files based on pre-defined criteria and manually inspect the FASTQC reports for the selected files only. Furthermore, based on the consolidated report, fastqc_parser extracts modules that showed a “FAIL” or “WARNING” messages for each filename. The result values for each failed module are then extracted individually for each filename and copied to the results folder.

To test the random_mer tool, datasets generated with the use of NEXTflex V2 kit (BIOO Scientific) were used. This small RNA library preparation kit contains four randomized bases that are included at the ligation junction on both the 3′ and 5′ adapters. The dataset information, adapters predicted by adapt_find, and randomized 4-mers predicted by the random_mer are given in Appendix A. Both adapt_find and random_mer tools accurately predicted adapters and randomized k-mers, respectively.

### 3.3. Output Files from Adapt_Find

The information on adapter sequence(s) and the type of adapter—along with the appropriate parameters—is passed on to cutadapt for adapter trimming. The trimmed adapter is placed under the folder named “good-mapping”. The log file from cutadapt is placed into “aux_files” folder and within “aux_files” under the corresponding filename subfolder. Similar to the query and subject FASTA files created for blasting, the BLAST output and the parsed output from BLAST are also placed in the corresponding filename subfolder. The trimmed file is then mapped against the user-provided reference genome using bowtie (if a corresponding bowtie index is provided), and the bowtie log file is parsed to extract the mapping percentage of trimmed reads. If the mapping percentage is greater than 50%, the trimmed FASTQ file is placed in the same folder (good-mapping). If the mapping percentage is less than 50%, it is moved to the “bad-mapping” folder.

Notably, a mapping percentage less than 50% does not necessarily indicate improper adapter trimming. Out of the 786 Illumina sequencing files on which the adapt_find have been tested, only three files had bad mapping after the adapter trimming. However, upon further manual inspection, the adapter sequences for files with bad mapping have been found to be correctly identified by adapt_find. In these cases, the percentage of reads with an adapter could be an indicator of proper adapter trimming. For example, dataset SRR953567 has only 47.41% mappable reads, but the percentage of reads with 3′ end adapter is 99.3% (Appendix A). Some of the sequencing files tested in this study contained much more reads than an average-sized sequencing file, yet the number of mappable reads was on a level of a typical file. Hence, the mapping percentage can also be related to the quality of the sequencing data rather than accuracy of the adapter prediction. Nevertheless, a manual inspection is recommended in such a case.

### 3.4. Comparison with Other Tools

We tested ten random publicly available single-read sequencing files downloaded from the Sequence Read Archive (SRA) database with different adapter types [14]. As shown in Table 1 and Appendix A, adapt_find accurately predicts adapter sequences, whereas DNApi, Minion and fastp tools are not always accurate. Sample output from fastp and adapt_find are shown in Appendix A, respectively. The 5′ and 3′ end adapter sequences from the top 21 most abundant sequences that were taken as query sequences are shown for dataset SRR578917 and SRR997335 in Appendix A, respectively. In addition, adapt_find was also tested on the 539 datasets used in the DNApi study [10]. adapt_find accurately identified adapter sequences in all the datasets (Appendix A), including those datasets where both DNApi and Minion inaccurately identified adapter sequences (n = 8 and n = 11, respectively). DNApi and adapt_find identified the correct adapter sequence in 35 libraries for which wrong adapter information was provided in the metadata (Appendix A).

### 3.5. Salient Features

Both adapt_find and randomer_find are implemented with a multi-processing module. Therefore, depending on the CPU capacity, the script can spawn processes for each input file and thereby parallelize the job, which in turn results in a considerably shorter execution time. Both tools can trim identified adapter sequences or random k-mers using cutadapt and check the mapping percentage after adapter trimming using bowtie, if a corresponding index is provided. adapt_find also outputs a comma separated value (csv) file, which contains the filename, the type of adapter, mapping percentage, and other relevant information (Appendix A). fastqc_parser can save a lot of time by automated extraction of information without the need for installing any packages or executing a batch of commands.

## 4. Discussion

adapt_find is precise and accurate in identifying adapter sequences from any type of raw single-read sequencing datasets. It performs better in terms of precise identification of adapters, when compared to fastp [8], Minion [9], and DNApi [10]. random_mer can identify random bases that are part of the library construction protocol in certain library preparation kits [4,15]. Both adapt_find and random_mer use multiprocessing modules by which multiple sequencing files are processed at the same time, provided there are adequate CPU resources. The fastqc_parser is a valuable addition to process multiple FASTQC reports. Pre-processing multiple raw sequencing files and the production of quality control reports for individual files are time consuming. adapt_find, randomer_find, and fastqc_parser significantly aid the process. These tools are relevant, as they facilitate big data analyses and re-use of rapidly growing genomic information.

The design of adapt_find algorithm reflects the data currently available in public repositories. This algorithm is based on three assumptions for the accurate detection of adapters. First, the algorithm of adapt_find requires the presence of intact adapters in raw sequencing data. Although this assumption may hold good for a major fraction of sequencing reads, in reality, adapter sequences may have insertions, deletions, or substitutions. This can affect the efficiency of analysis. There are four possible scenarios for the fraction of reads with modifications in adapter sequences: (1) fraction is relatively low; (2) modifications are frequent but homogenous (the same type of modification is present in majority of reads having modified adapter sequence); (3) modifications are frequent and heterogenous (diverse in size and type); and (4) all the reads have modified adapter sequences with diverse types and sites of modification. The original adapters would be identified under scenarios 1 or 3. In scenario 2, the modified adapter would still be predicted as an adapter and trimmed effectively using cutadapt. In the case of scenario 4, no adapter sequence would be identified. However, scenario 4 is hypothetical and from our experience in analyzing many datasets, we have never encountered such a situation. In all the datasets analyzed in this study, adapt_find could precisely identify the adapter sequence if there has been at least a minor fraction of reads with intact adapters. adapt_find was able to find the correct adapter sequences when the fraction of reads with intact adapters is as low as 1.6%, whereas DNApi predicted wrong adapter sequences in such datasets (n = 6) [10].

The second assumption is that the head part of the 3′ adapter sequence is intact. Hypothetically, if the fraction of reads with a degraded form of the head part of 3′ end adapter was substantial, the degraded 3′ end adapter sequence would be predicted as the adapter sequence. However, we have never come across such a dataset, and we have not encountered any report on the degraded form of the adapter at the head end. This assumption does not apply to the tail part of the 3′ end adapter, as cutadapt and most trimming tools only need a match at the head part of a 3′ end adapter.

The third assumption is that the bases at the end of the biological sequence preceding the adapter sequence are not the same for all the query and subject sequences; otherwise, the predicted adapter sequence would also contain these bases. However, this is a very rare case scenario. A similar issue that can result in the identification of a false positive adapter is when a certain biological sequence is over-represented in a particular library. DNApi identified false positive adapter sequences in two datasets where a particular biological sequence constituted 20–30% of all reads [10]. However, adapt_find accurately identified the correct adapter sequence in these two datasets (Appendix A). DNApi and Minion use k-mer approach, and fastp uses a seed tree method for identifying adapter sequences. While these methods can be useful in identifying adapter sequences, they can also call biological sequences as adapters. adapt_find overcomes this issue by collapsing all the reads and filtering alignments. Moreover, DNApi, Minion and fastp cannot identify 5′ end adapter sequences, whereas adapt_find can successfully identify both 5′ end and 3′ end adapters. In addition, unlike DNApi, adapt_find predicts whether the input reads are adapter-trimmed without mapping input reads to the genome. Furthermore, certain library construction protocols use random-mers on the 3′ end or both the 5′ and 3′ end of reads. A k-mer approach cannot be used to identify random-mers, while the random_mer tool identifies random-mers on either end of a read.

## 5. Conclusions

adapt_find is a tool for the high-throughput, automated identification of adapter sequences in single-read sequencing datasets, regardless of the type of the adapter and sequencing platform. It does not require any prior information about the adapters used. It can determine whether a dataset has already trimmed adapters or not, and it can identify adapter sequences with perfect accuracy. Furthermore, adapt_find summarizes the output in the form of a csv file with relevant information. random_mer is the tool to identify random “N” bases in sequencing files and it is accurate in predicting randomized adapters in sequencing files. fastqc_parser tool extracts quality check information from multiple FASTQC reports and consolidates them into a single report; also, it extracts specific results of the problematic modules from a QC report for sequencing files in a batch. These tools are simple and fully operable by a user with no background knowledge in bioinformatics. Together, they substantially facilitate the pre-processing and quality check of big batches of datasets prior to analyses.

## Figures and Tables

**Figure 1 biomolecules-10-00878-f001:**
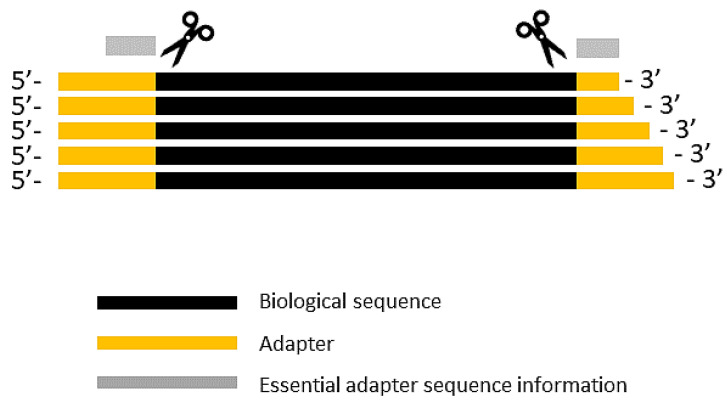
Essential information on adapter sequences for effective trimming process. The 5′ end adapter is usually of constant length, while the lengths of 3′ end adapters may vary in a dataset. The exact match to the 3′ end (tail part, adjacent to a biological sequence) of the 5′ end adapter, and to the 5′ end (head part, adjacent to a biological sequence) are required to identify the adapters. In the latter case, the shortest variant of the 3′ end adapter suffices.

**Figure 2 biomolecules-10-00878-f002:**
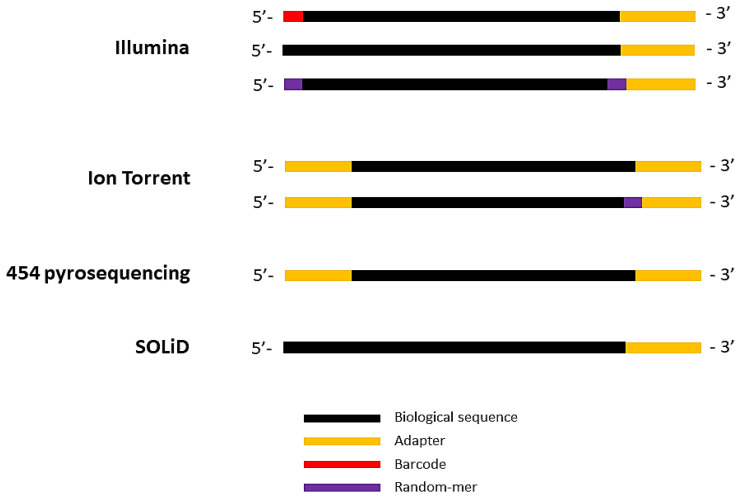
Schematic representation of output reads data format in different sequencing technologies. In all the four sequencing technologies, 3′ end adapters are ligated to biological sequences in the sequencing outputs; in addition, 5′ end adapters are present in Ion Torrent and 454 pyrosequencing outputs. The Illumina output reads may have four-letter barcode in the 5′ end, and/or random 4 “N” nucleotides at both ends. Similarly, depending on the library preparation kit used, output reads from Ion Torrent might have random 5-mer and a three-letter barcode in addition to 5′ end and 3′ end adapters.

**Figure 3 biomolecules-10-00878-f003:**
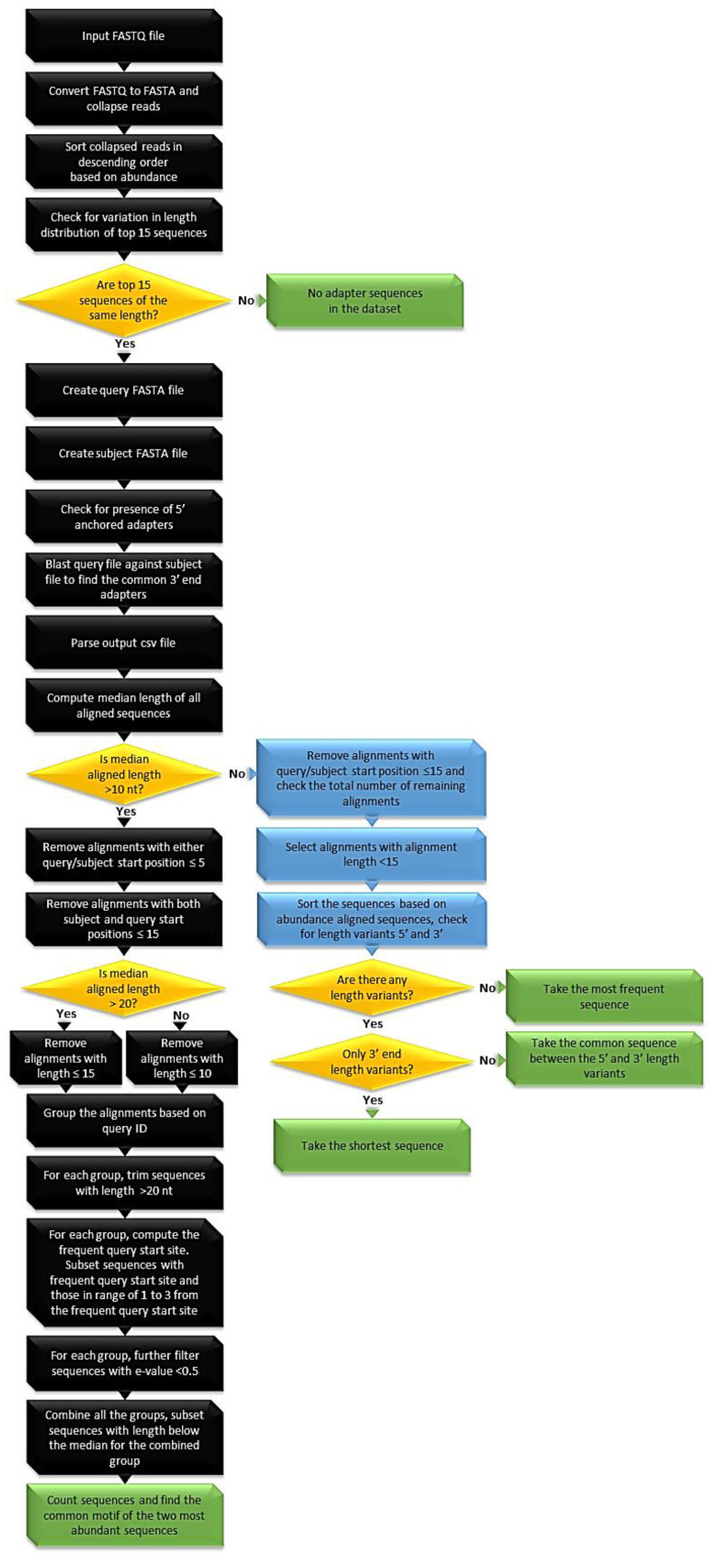
The adapt_find workflow. Black boxes: general procedure, green boxes: exit step, blue boxes: alternative strategy, yellow diamonds: decision.

**Figure 4 biomolecules-10-00878-f004:**
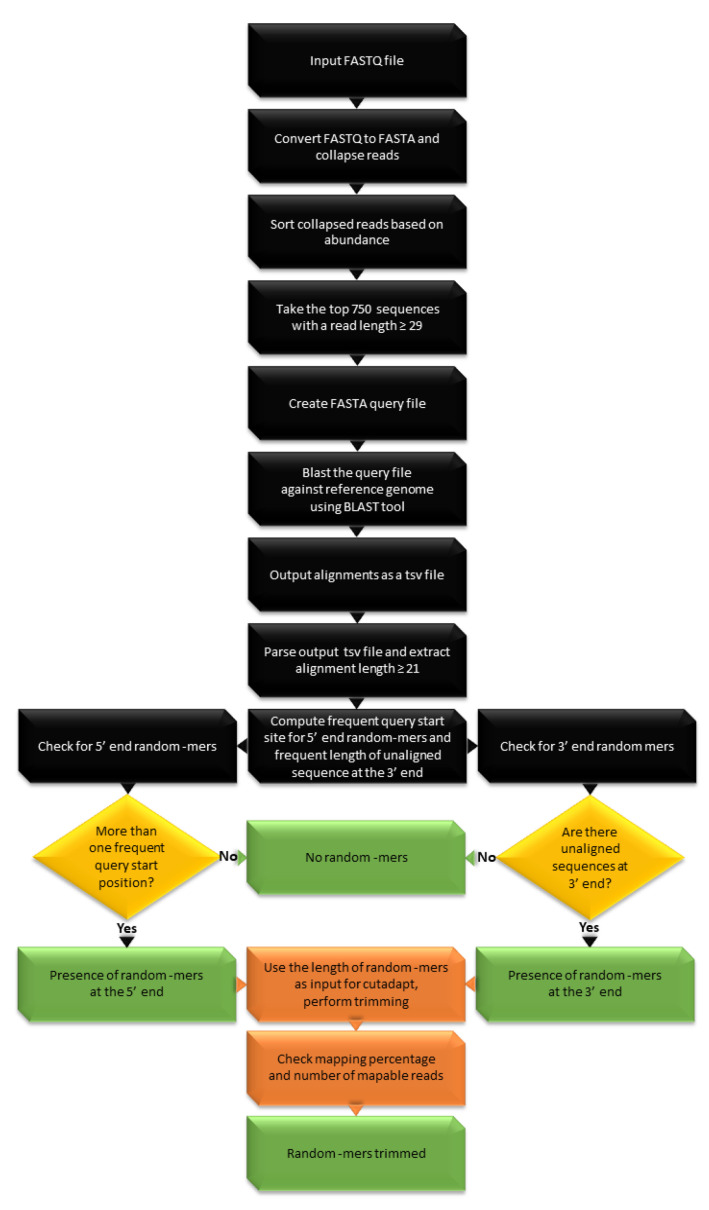
random_mer workflow. Black boxes: general procedure, green boxes: exit step, orange boxes: further recommended process, yellow diamonds: decision.

**Table 1 biomolecules-10-00878-t001:** Adapter sequences in ten randomly chosen datasets predicted using Adapt_Find (default parameters used). “Original Sequence” is the adapter sequence obtained from the Methods section of the original study; “na”—no adapter applied in the dataset and/or no adapter identified by an algorithm.

Dataset	Adapter	Original Sequence	Adapt_Find
SRR3151643	5′	na	na
	3′	AGATCGGAAGAGCACACGTCT	AGATCGGAAGAGCACACGTC
SRR578915	5′	ACTA	ACTA
	3′	TCGTATG	TCGTATG
SRR578914	5′	ACTA	ACTA
	3′	TCGTATG	TCGTATG
SRR578911	5′	ATCC	ATCC
	3′	TCGTATG	TCGTATG
SRR578918	5′	ATCC	ATCC
	3′	TCGTATG	TCGTATG
SRR578913	5′	na	na
	3′	TCGTATG	TCGTATG
SRR5122167	5′	na	na
	3′	na	na
SRR1427469	5′	na	na
	3′	na	na
SRR997332	5′	na	na
	3′	TGGAATTCTCGGGTGC	TGGAATTCTCGGGTGC
SRR953574	5′	na	na
	3′	CTGTAGGCACCAT	CTGTAGGCACCATC

**Table 2 biomolecules-10-00878-t002:** adapt_find corrects the source information on adapter sequences. The discrepancies between the 3′ end adapter sequence provided in the original study and the sequences predicted by adapt_find were manually inspected. The “grep” command revealed the frequent presence of the adapters identified by adapt_find in the raw sequencing files, while the sequence given in the source study was not found in the sequencing dataset. Also, adapter trimming using the parameters specified in the original study with the use of cutadapt was not successful.

Dataset	Adapter	Original Study	Adapt_ Find
SRR3225657	5′	ATTGATGGTGCCTACAG	ATTGATGGTGCCTACAGA
	3′	GATCGTTCGGACTGTAGATC	AGTGATCGTCGGACTGTAG
SRR3225658	5′	ATTGATGGTGCCTACAG	ATTGATGGTGCCTACAGA
	3′	GATCGTTCGGACTGTAGATC	TTCGATCGTCGGACTGTAG
SRR3225659	5′	ATTGATGGTGCCTACAG	ATTGATGGTGCCTACAGA
	3′	GATCGTTCGGACTGTAGATC	CTTGATCGTCGGACTGTAG
SRR3225660	5′	ATTGATGGTGCCTACAG	ATTGATGGTGCCTACAGA
	3′	GATCGTTCGGACTGTAGATC	CCCGATCGTCGGACTGTAG
SRR3225661	5′	ATTGATGGTGCCTACAG	ATTGATGGTGCCTACAGA
	3′	GATCGTTCGGACTGTAGATC	TCTGATCGTCGGACTGTAG
SRR3225662	5′	ATTGATGGTGCCTACAG	ATTGATGGTGCCTACAGA
	3′	GATCGTTCGGACTGTAGATC	TAAGATCGTCGGACTGTAG

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
