# Peer review of "High-Throughput Identification of Adapters in Single-Read Sequencing Data"

_biomolecules, 2020, doi:10.3390/biom10060878_

Round 1
Reviewer 1 Report
In the manuscript by Mohideen et al., the authors describe an automated tool to computationally detect and annotate adaptors in multiple datasets without any previous knowledge. Whilst the removal of adaptors is relatively trivial for experienced researchers, it is indeed a significant problem in meta studies using large amounts of publicly available data. The tool described by Mohideen et al., would be a useful contribution, and in their metrics it performs better than previous tools.
My review of the paper was however severely hampered due to a problem with the website hosting the code, which was offline. This meant that I could not test the software myself. Additionally the software is implemented in python 2, which is about to become obsolete. Consequently, due to these problems I cannot recommend publication of this manuscript in its current form, unless the authors can revise.
Major Issues:
1. The domain name mentioned in the paper reports as expired on 2020-3-6 (checked on 2020-4-10). I strongly recommend the authors to upload their code to a more permanent website, such as GitHub, sourceforge, or bitbucket, and make the code available.
2. adapt_find is written in python 2.7. The python 2 series reaches end of life in 2020, and will become obsolete after that. The tool should be ported to Python 3. Without this, it will quickly become unusable as most users of python have or will very soon migrate to python 3. Indeed, due to the problem with the website, and the fact that I no longer maintain a python 2 install, means that I am unable to test this tool on my own data. Porting to python 3 is relatively straightforward, and there is no reason not to do it.
Minor comments:
3. The license for the software is not mentioned. Open source, commercial, etc.
4. I would like the authors to comment on the possibility that their approach can detect genome repeats or not. I think the authors BLAST the resulting putative adaptors against the genome. However, the writing is not clear on how this is dealt with.
5. Table 1: Whilst useful, I find this not systematic enough. All of the randomly selected samples are from C.elegans or Danio Rerio. A remote possibility is that genome repeats might be picked up in other species genomes and recognized as adaptors (also point 3). For example C. Elegans is predicted to have ~10% genome repearts, whereas D.rerio and human is more like 40%:
http://www.repeatmasker.org/species/danRer.html
http://www.repeatmasker.org/species/ce.html
http://www.repeatmasker.org/species/hg.html
Importantly the types of repeat are quite different between species. Do the authors see any repeats coming up as false+ adaptors? This seems unlikely as the authors enforce the adaptors to the 5’ and 3’ ends, but nonetheless it would be useful to rule it out.
6. For the 800+ datasets the authors interrogated. Did they observe any false+ adaptors?
Author Response
We wish to thank the Reviewer very much for his/her constructive comments and help in improving the manuscript. We would like to apologise the link to the tool we provided has not been working at the time of review. We have addressed all the points and fixed all the problems. Please find the detailed responses below.
Comments and Suggestions for Authors
In the manuscript by Mohideen et al., the authors describe an automated tool to computationally detect and annotate adaptors in multiple datasets without any previous knowledge. Whilst the removal of adaptors is relatively trivial for experienced researchers, it is indeed a significant problem in meta studies using large amounts of publicly available data. The tool described by Mohideen et al., would be a useful contribution, and in their metrics it performs better than previous tools.
My review of the paper was however severely hampered due to a problem with the website hosting the code, which was offline. This meant that I could not test the software myself. Additionally the software is implemented in python 2, which is about to become obsolete. Consequently, due to these problems I cannot recommend publication of this manuscript in its current form, unless the authors can revise.
Major Issues:
- The domain name mentioned in the paper reports as expired on 2020-3-6 (checked on 2020-4-10). I strongly recommend the authors to upload their code to a more permanent website, such as GitHub, sourceforge, or bitbucket, and make the code available.
RESPONSE: We are very sorry for this. We had unexpected problems with the server maintenance due to COVID-19 outbreak. We have uploaded the code in GitHub at https://github.com/asan-nasa/adapt_find
- adapt_find is written in python 2.7. The python 2 series reaches end of life in 2020, and will become obsolete after that. The tool should be ported to Python 3. Without this, it will quickly become unusable as most users of python have or will very soon migrate to python 3. Indeed, due to the problem with the website, and the fact that I no longer maintain a python 2 install, means that I am unable to test this tool on my own data. Porting to python 3 is relatively straightforward, and there is no reason not to do it.
RESPONSE: Thank you for the valuable suggestion. It is now ported to PYTHON 3.6.8, also compatible with PYTHON 2.7.5.
Minor comments:
- The license for the software is not mentioned. Open source, commercial, etc.
RESPONSE: MIT License (similar to open source: use, copy, modify, merge, publish, distribute, sublicense, and/or sell copies of the Software). Please find this information on page 14, line 453.
The permissions, limitations and conditions of the license are listed here: https://github.com/asan-nasa/adapt_find/blob/master/LICENSE
- I would like the authors to comment on the possibility that their approach can detect genome repeats or not. I think the authors BLAST the resulting putative adaptors against the genome. However, the writing is not clear on how this is dealt with.
RESPONSE: We have re-phrased the information to be clearer. The putative adapters are not blasted against the genome. adapt_find can detect adapters from single end sequencing reads. The add-on tool random_mer can detect random N-mers on either end of the adapter-trimmed read sequences. In adapt_find, both query and subject sequences for BLAST are selected from collapsed raw reads (please see Page 4, lines 142-4). In random_mer, the query sequences are selected from collapsed raw reads and the subject FASTA is the reference genome (please find this information on Page 8, lines 257-60). In adapt_find, adapter-trimmed fastq files can be mapped to the genome afterwords, if a corresponding bowtie index is provided (please see page10, lines 316-8). However, both tools can identify only adapters, not genome repeats. Most of the genome repeats are low complexity sequences and adapt_find removes low complexity sequences while selecting query and subject sequences (please find this information on page 4, lines 144-6). The remaining genome repeats are removed while filtering biological sequences (page 6).
- Table 1: Whilst useful, I find this not systematic enough. All of the randomly selected samples are from C.elegans or Danio Rerio. A remote possibility is that genome repeats might be picked up in other species genomes and recognized as adaptors (also point 3). For example C. Elegans is predicted to have ~10% genome repearts, whereas D.rerio and human is more like 40%:
http://www.repeatmasker.org/species/danRer.html
http://www.repeatmasker.org/species/ce.html
http://www.repeatmasker.org/species/hg.html
Importantly the types of repeat are quite different between species. Do the authors see any repeats coming up as false+ adaptors? This seems unlikely as the authors enforce the adaptors to the 5’ and 3’ ends, but nonetheless it would be useful to rule it out.
RESPONSE: Although the small RNA datasets used as test datasets come from different species, the main rationale is not to include samples from different species, but to include datasets with different library construction protocols. In Table 1, there are three different types of datasets: 1) without adapters, 2) those having both 5’ and 3’ adapters, and 3) those having 3’ adapters only. These datasets are small RNA datasets from different species from different biological conditions. It is not known how many small RNAs are derived from genome repeats in these datasets. However, we did not identify any genome repeats as false+ adaptors. The main reason is that the raw read sequences are collapsed and for the blasting, they are considered as single sequence (please see the revised section 2.3.)
- For the 800+ datasets the authors interrogated. Did they observe any false+ adaptors?
RESPONSE: We didn’t find any false positive adapters in any dataset examined (please see the section 3.2). Moreover, adapt_find has been able to identify adapters correctly in the datasets, where incorrect adapters were reported (please see Table 2).
Reviewer 2 Report
I think that the focus of the article is not entirely correct due to several erroneous assertions by the authors.
First: There are tools to predict adapter sequences: Minion (in the kraken package), adapterRemoval (version 1 and 2) and DNApi. At least the first and the last, work with single and paired reads. You should read these the two last papers and see how they compare their tools against minion.
In fact the miARma-seq pipeline includes Minion to automatically predict the adapter sequence and uses cutadapt to remove these sequences. This pipelien also executes fastqc before and after this process to compare the differences in quality terms from this process.
Then, in a default analytical pipeline, these sequences are not always removed. First, a quality study is done and if the accumulation of these sequences is small (quite usual in an RNASeq), then it is not necessary to remove them since it does not affect the results.
In fact, most authors only recommend removing adapter sequences in small RNAseq.
Finally, I would recommend you to update the program to python 3.x because python 2.x is no longer supported
Author Response
We wish to thank the Reviewer very much for his/her constructive comments and help in improving the manuscript. We have addressed all the points and performed additional analyses as suggested. Please find the detailed responses below.
Comments and Suggestions for Authors
I think that the focus of the article is not entirely correct due to several erroneous assertions by the authors.
First: There are tools to predict adapter sequences: Minion (in the kraken package), adapterRemoval (version 1 and 2) and DNApi. At least the first and the last, work with single and paired reads. You should read these the two last papers and see how they compare their tools against minion.
In fact the miARma-seq pipeline includes Minion to automatically predict the adapter sequence and uses cutadapt to remove these sequences. This pipeline also executes fastqc before and after this process to compare the differences in quality terms from this process.
RESPONSE: Thank you for suggesting these tools for the comparison. In the revised version, comparisons were made between adapt_find, fastp, Minion and DNApi, please see the new Additional file 13. In addition, adapt_find was also tested on the 539 datasets used in the DNApi study, please see the new Additional file 14. The comparisons show the superiority of adapt_find and are addressed in the revised section 3.4. on Page 12, and Discussion on Page 13.
adapterRemoval (versions 1 and 2) only trims adapter sequences. It does not identify adapter sequences (https://github.com/MikkelSchubert/adapterremoval). Thus, it cannot be compared to adapt_find.
Then, in a default analytical pipeline, these sequences are not always removed. First, a quality study is done and if the accumulation of these sequences is small (quite usual in an RNASeq), then it is not necessary to remove them since it does not affect the results.
In fact, most authors only recommend removing adapter sequences in small RNAseq.
RESPONSE: In a typical small RNA sequencing file, % of reads with adapter sequences is as high as 100% for 5’ end adapters and approx. 60 to 95% in case of 3’ end adapters. Most datasets uploaded to public databases have raw reads with adapters. Actually, adapt_find checks whether the input fastq files are adapter-trimmed or not (Please see Figure 3). adapt_find does not perform adapter prediction and trimming if the input fastq files are already adapter-trimmed. While the other tools: Minion and DNApi, do not check whether the input files are adapter-trimmed by default (Please see Additional file 13). DNApi maps the raw reads and adapter (predicted) trimmed fastq to the genome and compares the mapping percentage (raw vs adapter-trimmed) to come to a conclusion whether the input files are adapter-trimmed fastq files or not. Whereas, adapt_find can predict whether the input files are adapter trimmed or not without mapping it to a genome. Please find this question discussed on Page 2, lines 64-7, and in Discussion (mostly Page 13).
Finally, I would recommend you to update the program to python 3.x because python 2.x is no longer supported
RESPONSE: Thank you for the valuable suggestion. It is now updated to PYTHON 3.6.8, also compatible with PYTHON 2.7.5.
Round 2
Reviewer 1 Report
The authors have revised the text, and I am satisfied. However, there are several bugs in the adapt_find script that need to be fixed: 1. The multiprocessing parts needs to have the executable script part enclosed in a if __name__ == "__main__":, otherwise it recursively calls itself (it throws an error on my python 3 environment). This needs to be added at line ~933, and everything below should be tabbed into this segment. 2. adapt_find looks for blastn version 2.7.1, but this is not the latest version (2.10.1), and anyway you should just test for its presence and necessary features, not the specific version. 3. the script attempts to install cutadapt using pip. But the user may not desire this if they are (for example) using a conda environment. In my opinion in these cases the script should just exit with an error and ask the user to install bowtie, cutadapt and blastn, rather than attempting to install through the script. Depending upon the users environment, a custom install could be a bad idea. 4. The script should support gzipped FASTQ files.Author Response
We wish to thank the Reviewer very much for spotting the bugs and helpful suggestions on the organization of the script. We have addressed all the points and fixed the bugs. Please find the detailed responses below.
Comments and Suggestions for Authors
The authors have revised the text, and I am satisfied. However, there are several bugs in the adapt_find script that need to be fixed:
- The multiprocessing parts needs to have the executable script part enclosed in a if __name__ == "__main__":, otherwise it recursively calls itself (it throws an error on my python 3 environment). This needs to be added at line ~933, and everything below should be tabbed into this segment.
RESPONSE: Thank you very much.The is fixed now. The changes can be found in the following commit: https://github.com/asan-nasa/adapt_find/commit/1dd752cb49263e881611b71b7d099825301c1d45
- adapt_find looks for blastn version 2.7.1, but this is not the latest version (2.10.1), and anyway you should just test for its presence and necessary features, not the specific version. 3. the script attempts to install cutadapt using pip. But the user may not desire this if they are (for example) using a conda environment. In my opinion in these cases the script should just exit with an error and ask the user to install bowtie, cutadapt and blastn, rather than attempting to install through the script. Depending upon the users environment, a custom install could be a bad idea.
RESPONSE: Thank for the suggestions. We have made changes accordingly. Now the script checks whether bowtie, cutadapt and blastn are available. It does not attempt to install, instead, it exits with an error asking users to install bowtie, cutadapt or blastn. The changes can be found in the following commit: https://github.com/asan-nasa/adapt_find/commit/d376a0e5d60c7f037ae5c8eae2843efcaf54e321
Changes made to the text in the manuscript can be found in page number 2, line number 79 (“or they would be installed locally” is removed from the sentence)
Also, adapt_find is tested with the latest version of blastn (2.10.1), cutadapt (2.10) and bowtie (1.2.3).
- The script should support gzipped FASTQ files.
RESPONSE: Changes are made to the script such that the script can read input gzipped FASTQ files. Changes can found under the following commit: https://github.com/asan-nasa/adapt_find/commit/0f81ceeb62c6456fd9e4f6934c39cc8c491528b5
Reviewer 2 Report
I would like to thank the authors for the effort made to answer my questions. I think the article has improved a lot in this review. The authors talk about their tool, they compare it with other similar tools demonstrating in what they are better at. I think with that the article has improved enough to be accepted.
But, It is true that I see it necessary to modify some parts of the text. For example:
a) It is not seem elegant to write in the abstract that there are tools to eliminate adapter sequences but that these are not reliable. Better to say that "they have certain limitations"
b) I think it is fine to point out that adapt_find works with sequences from different sequencers (what about PacBio?), but it is not so important because 99% are sequenced with Illumina. However, I do think that adapt_find, for me, has a very important point and I don't see it highlighted. It is the fact of detecting if your files already have the adapter removed. This is something that seems very useful to me because in a pipeline you can have this program always activated and if there is an adapter it is removed and if not, then you automatically move to the next step.
Author Response
We wish to thank the Reviewer very much for his/her constructive comments and help in improving the manuscript. Please find the detailed responses below.
Comments and Suggestions for Authors
I would like to thank the authors for the effort made to answer my questions. I think the article has improved a lot in this review. The authors talk about their tool, they compare it with other similar tools demonstrating in what they are better at. I think with that the article has improved enough to be accepted.
But, It is true that I see it necessary to modify some parts of the text. For example:
- a) It is not seem elegant to write in the abstract that there are tools to eliminate adapter sequences but that these are not reliable. Better to say that "they have certain limitations"
RESPONSE: Thank you for this suggestion. We have corrected the text accodingly (Page 1, line 13).
- b) I think it is fine to point out that adapt_find works with sequences from different sequencers (what about PacBio?), but it is not so important because 99% are sequenced with Illumina. However, I do think that adapt_find, for me, has a very important point and I don't see it highlighted. It is the fact of detecting if your files already have the adapter removed. This is something that seems very useful to me because in a pipeline you can have this program always activated and if there is an adapter it is removed and if not, then you automatically move to the next step.
RESPONSE: We found no small RNA dataset in SRA which has been sequenced using Pacbio. Perhaps because Pacbio is used only for sequencing long reads, typically greater than 10000 bp (https://www.sciencedirect.com/science/article/pii/S1672022915001345 and https://genomebiology.biomedcentral.com/articles/10.1186/s13059-020-1935-5 ). The Single Molecule, Real-Time (SMRT) Sequencing of Pacbio is most suited for applications that require long read length and we could not find any specific kit that Pacbio uses for short reads (https://www.pacb.com/applications/).
Thank you for highlighting the feature of adapt_find related to its ability to determine whether the adapters were already trimmed or not. Apart from the information in Page 4, lines 134-6, and Page 13, lines 417-9, we have highlighted it in Conclusions (Page 13, lines 425-6).